# Breast Cancer Screening Using Inverse Modeling of Surface Temperatures and Steady-State Thermal Imaging

**DOI:** 10.3390/cancers16122264

**Published:** 2024-06-19

**Authors:** Nithya Sritharan, Carlos Gutierrez, Isaac Perez-Raya, Jose-Luis Gonzalez-Hernandez, Alyssa Owens, Donnette Dabydeen, Lori Medeiros, Satish Kandlikar, Pradyumna Phatak

**Affiliations:** 1Department of Hematology-Oncology, Rochester Regional Health, Rochester, NY 14621, USA; nithya.sritharan@rochesterregional.org (N.S.); donnette.dabydeen@rochesterregional.org (D.D.); lori.medeiros@rochesterregional.org (L.M.); 2Department of Mechanical Engineering, Rochester Institute of Technology, Rochester, NY 14623, USA; cg9804@rit.edu (C.G.); iperez@biredimaging.com (I.P.-R.); jlgh1188@gmail.com (J.-L.G.-H.); anr493@gmail.com (A.O.); skandlikar@biredimaging.com (S.K.); 3BiRed Imaging Inc., Rochester, NY 14609, USA

**Keywords:** inverse modeling, breast cancer screening, heat generation map, physics-informed neural network

## Abstract

**Simple Summary:**

Early cancer detection is crucial for favorable patient outcomes, with mammography playing a central role in breast cancer detection. However, challenges persist, such as limited sensitivity in dense breast tissue and low specificity leading to excessive invasive testing. Leveraging the distinct biological characteristics of malignant tumors, we employed high-sensitivity thermal imaging to identify temperature changes associated with cancer. This innovative approach integrates a novel imaging technique and a physics-based prediction model to accurately ascertain the presence of breast cancer, regardless of size, location, and breast density. Our goal is to develop this technique as a complementary tool to mammography for widespread screening and as a cost-effective, stand-alone method for specific populations underserved by mammography.

**Abstract:**

Cancer is characterized by increased metabolic activity and vascularity, leading to temperature changes in cancerous tissues compared to normal cells. This study focused on patients with abnormal mammogram findings or a clinical suspicion of breast cancer, exclusively those confirmed by biopsy. Utilizing an ultra-high sensitivity thermal camera and prone patient positioning, we measured surface temperatures integrated with an inverse modeling technique based on heat transfer principles to predict malignant breast lesions. Involving 25 breast tumors, our technique accurately predicted all tumors, with maximum errors below 5 mm in size and less than 1 cm in tumor location. Predictive efficacy was unaffected by tumor size, location, or breast density, with no aberrant predictions in the contralateral normal breast. Infrared temperature profiles and inverse modeling using both techniques successfully predicted breast cancer, highlighting its potential in breast cancer screening.

## 1. Introduction

Breast cancer is the most prevalent cancer affecting women in the United States and stands as the second-leading cause of death among them [1]. While it predominantly impacts women aged 50 years or older, presenting at a median age of 62 years [2], breast cancer also poses challenges for younger women, often manifesting at advanced stages. Mammogram screening exhibits limited sensitivity in younger women, stemming from challenges posed by dense breast tissue that conceals underlying lesions and in cases of lobular carcinomas where discrete tumors are often not readily apparent [3,4]. It is estimated that more than 40% of women have dense breast tissue, which are four to six times more at risk of developing cancer [4,5,6,7]. Despite the advancements offered by 3D digital breast mammograms to address this issue, they come with increased radiation exposure [8,9]. Various adjunctive methods, including ultrasound, magnetic resonance imaging (MRI), and contrast-enhanced mammography, have shown promise in enhancing breast cancer detection. Computer aided detection (CAD) programs have been developed to aid and improve mammography, utilizing tools such as artificial intelligence and other computation techniques [10,11,12,13,14,15]. However, these methods also present challenges, leading to high false positives and unnecessary invasive testing, with less than 1% of these women eventually diagnosed with breast cancer [16,17,18,19]. Consequently, there is a persistent demand for cost-effective screening methods, both as complements to mammography and as potential stand-alone techniques, especially for specific patient populations not optimally served with mammography, with a simultaneous focus on minimizing cumulative radiation exposure.

### Thermal Imaging Background

Thermal imaging is a method that captures the infrared radiation emitted from the body captured by an infrared camera as surface temperature measurements [20]. Strąkowska et al. [21] has shown the ability of utilizing thermal imaging in the detection of skin diseases. In 1956, Lawson [22,23,24,25] showed that malignant tumors generated an increase in temperature measurements that can be captured through thermal imaging. Our research endeavors in thermal imaging for breast cancer detection have been ongoing since 2015 [26,27,28,29,30,31,32], with a focus on overcoming the limitations of traditional approaches. Malignant tumors are characterized by increased angiogenesis and metabolic rates to sustain rapid cell proliferation [33,34,35]. Traditional thermal imaging techniques have historically faced challenges in detecting malignant tumors, with contributing factors including low-sensitivity cameras unable to detect subtle temperature changes, the influence of body temperature transmitted through the chest wall, and a lack of a clear methodology for translating identified hot spots on breast surfaces [30,31,32]. To address these issues, we developed a novel steady-state prone position imaging method, eliminating the impact of chest wall motion and variability in breast temperatures associated with imaging in the supine position. The relation between the emissive power of a body and its temperature is based on the Stefan–Boltzmann Law and, in the human body, is noted to be independent of skin color [36,37,38]. This characteristic can be used to obtain the surface temperature of breasts as previously described by Lahiri et al. [39] and to detect abnormalities that would alter this temperature distribution. Gautherie [40,41,42] correlated thermal conductivity and surface temperature measurements to increased vasculatures and metabolic activity in tumor tissues. He developed a model that encapsulated the thermal effect from the metabolic activity as a heat source dependent on the tumor volume and doubling time [43,44,45]. This allowed researchers to model the tumor as a heat source dependent on the tumor size. We developed an algorithm leveraging the well-established Pennes’ bioheat transfer model [46,47,48,49,50], which helps accurately predict hot spots located deep within breast tissue using surface temperature measurements, along with 3D modeling of the breasts using MRI images. 

By enhancing the precision of thermal imaging in breast cancer diagnosis, we aspire to contribute significantly to early detection efforts and ultimately improve patient outcomes.

## 2. Materials and Methods

We conducted a prospective, non-invasive study at Rochester General Hospital (RGH) between March 2018 and September 2019, after approval by the Institutional Review Board (IRB) at RGH. This study was conducted in collaboration with researchers at the Rochester Institute of Technology (RIT).

Inclusion criteria incorporated all of the following:Women 21 years or older;Able to provide informed consent;Breast lesions identified as BIRADS 4 or BIRADS 5, or detected clinically on physical exam;Malignancy confirmed by biopsy of the lesion.

A total of 30 women were recruited to undergo the study protocol. Informed consent was obtained from the subjects regarding the procedure and effects of infrared imaging.

MRI images of bilateral breasts of the patients were captured in the prone position with a GE 3T MRI scanner (GE Healthcare, Chicago, IL, USA) under an IRB-approved protocol prior to biopsy [51]. Axial, coronal, and sagittal imaging were captured pre- and post-contrast with GE 3T MRI. Multi-view infrared images, which included 8 viewed at 45° intervals around the bilateral breasts, were captured in the same prone position as the MRI, using an FLIR (Wilsonville, OR, USA) SC6700 IR camera with a thermal sensitivity of 0.02 °C. Capturing infrared images of the breast in the prone position allowed for a full 360° view, with each breast being imaged and analyzed separately for each patient. Figure 1 shows an illustration of the infrared imaging procedure and the multi-view infrared data of a patient [52]. The patients were imaged after about 10 min of acclimation to the room to ensure the body reached a steady-state temperature.

The following steps were conducted for each patient by the RIT researchers with the obtained images, with no transmission of patient identifiers:Patient-specific digital breast models of both breasts generated using MRI images through image processing and 3D reconstruction techniques with ImageJ 1.51n software as described in Gonzalez-Hernandez et al. [53];Simulated breast surface temperatures of each breast generated using the patient-specific breast model with Ansys Fluent 2019 R2 thermal modeling software for thermal modeling of breast cancer;Tumor size and location predicted through inverse modeling using the Levenberg–Marquard algorithm (LMA) for inverse modeling [54].Infrared imaging data integrated with the above software and algorithm to accurately predict the presence or absence of breast cancer.

Figure 2 shows a process flowchart for detecting the presence or absence of breast cancer using infrared imaging, thermal modeling, and inverse modeling. This process relies on three primary inputs: (i) initial tumor parameters, including size and location, (ii) a patient-specific breast model, and (iii) infrared temperatures from multi-view images. Initial tumor parameters, set to a diameter of 1.8 cm and placed centrally within the breast geometry, along with the patient-specific breast model, are fed into the thermal modeling software to generate simulated surface temperatures. These simulated temperatures, alongside actual IR temperatures, are inputted simultaneously into the LMA detection algorithm for alignment and comparison. If disparities arise between simulated and actual temperatures, the algorithm iterates, updating tumor parameters until a best match is achieved, signaling either tumor detection or its absence. In the case of detection, the algorithm predicts tumor size and location within the breast. Conversely, if no tumor is detected, the algorithm positions it in areas minimally impacting surface temperature, such as outside the breast or at the chest wall, as illustrated in Figure 2. This iterative process is applied to each patient and each breast individually. 

Gonzalez-Hernandez et al. [54] provide a detailed account of the inverse modeling process employed for breast cancer detection using surface temperatures. This methodology underwent initial validation on seven biopsy-proven breast cancer patients, detailed in both Gonzalez-Hernandez et al. [54] and Recinella et al. [51]. Gutierrez et al. [52] subsequently automated the algorithm to accommodate a larger volume of cases, implementing enhancements to optimize its runtime. In this investigation, ANSYS Fluent and the LMA algorithm were utilized for patients with accessible MRI data.

Clinical data, encompassing patient age, breast tissue density, tumor location, breast tissue composition evaluated via mammography, and tumor histology, were extracted from the electronic medical record (EMR) and deidentified prior to analysis. All patients included in this study underwent surgical resection of tumors with curative intent. Surgical pathology findings were categorized based on histological types, tumor grade, and estrogen receptor (ER) and progesterone receptor (PR) expression, as well as Her2/neu status. The actual tumor diameter was approximated by the volume calculated from MRI image dimensions. Statistical analysis using ANOVA was conducted to compare estimated and actual tumor sizes. Additionally, all enrolled patients received clinical follow-up for a minimum of two years to monitor for any instances of recurrent breast cancer in close proximity to the initial tumors. In this work, the positive control was breasts with verified malignancy, while the negative control was the contralateral healthy breasts.

## 3. Results

The final analysis included 24 patients diagnosed with breast tumors who underwent both MRI and infrared imaging, yielding a total of 25 breast tumors and 23 contralateral breasts devoid of tumors. Demographic and tumor specifics are outlined for each individual patient in Table 1. 

All patients were biologically female, aged between 42 and 72 years, with a median age of 67 years. Among them, 12 patients presented with left-sided tumors, 11 presented with right-sided tumors, and 1 patient exhibited bilateral breast tumors. Breast tissue density was classified into categories: predominantly fatty (PF), scattered fibro-glandular (SF), heterogeneously dense (HD), and extremely dense tissue (ED). The majority of patients had fibro-glandular breast tissue, with 5 patients having heterogeneously dense breasts and 1 with extremely dense breast tissue. Quadrant division delineated breast tissue involvement, specifically in the upper outer (UOQ), upper inner (UIQ), lower outer (LOQ), and lower inner (LIQ) quadrants, with most tumors located in the UOQ and fewest in the LIQ, consistent with epidemiological expectations. Histological analysis revealed various tumor types including atypical ductal hyperplasia (ADH), ductal carcinoma in situ (DCIS), invasive ductal carcinoma (IDC), invasive lobular carcinoma (ILC), and lobular carcinoma in situ, with tumor grades ranging from 1 to 3. Tumors were further examined for the presence of estrogen (ER), progesterone (PR), and HER2 receptors.

The algorithm demonstrated the successful detection of all biopsy-confirmed breast cancers, as depicted in Figure 3. Furthermore, none of the 23 contralateral breasts exhibited predicted tumors. Accurate estimation of tumor sizes was achieved, with a maximum error of 7 mm observed between actual and estimated tumor sizes, accompanied by variability in size differences. The sensitivity of tumor detection extended to tumors as small as 5 mm. A trend towards larger estimated thermal signatures compared to actual tumor sizes was noted in high-grade tumors (*p* = 0.181), as well as in cases with concomitant in situ carcinoma and invasive carcinoma (*p* = 0.466) and triple-negative breast cancer (*p* = 0.784).

## 4. Discussion

Breast cancer screening has played a pivotal role in reducing mortality rates associated with the disease by facilitating early tumor detection and improving cure rates. Despite the central role of mammograms in routine screening, challenges such as discomfort during procedures, radiation exposure, and limited efficacy in dense breast tissue persist. To address these challenges, various adjunct techniques have been explored. In our study, we leveraged infrared technology, extensively researched for decades, in combination with advanced computational fluid dynamics software. This innovative approach yielded significant success in accurately predicting the presence of cancerous tumors.

Our algorithm leverages the principle that metabolically active, well-perfused malignant tumors serve as heat sources, influencing breast surface temperature. It reliably and accurately predicted breast cancer regardless of tumor location, tissue density, histological subtype, prognostic receptor status, or tumor depth by analyzing breast surface temperature data. This comprehensive predictive capability extends to precancerous lesions, including atypical ductal hyperplasia. The patient cohort encompassed commonly encountered histological subtypes such as IDC, ILC, DCIS, and LCIS, confirming the technique’s broad applicability. The precision in predicting tumor sizes, with errors consistently below 1 cm, further highlights the robustness of our approach. Further studies on the minimal tumor size and other factors that affect the detectability of breast cancer using our algorithm are in development. Gutierrez and Kandlikar [55] have identified the detectability of the algorithm through numerical studies.

The observed correlation between thermal signatures and tumor characteristics aligns with known attributes of malignant tumors, characterized by heightened metabolic rates and increased vascularity. Notably, our method identified larger thermal signatures in high-grade and triple-negative tumors, both known for their rapid growth and expected elevation in angiogenesis and metabolism. Although the small sample size may limit the statistical significance of these findings, the observed trends hint at the potential to gain valuable insights into tumor behavior even before biopsy, thereby influencing patient care decisions.

The absence of false positives in contralateral breasts enhances the credibility of our method for detecting malignant lesions. Encouraged by these promising outcomes, our next step involves expanding our investigations through larger clinical studies. This will enable a more comprehensive evaluation of the sensitivity and specificity of infrared imaging using physics-based inverse modeling with breast surface temperatures as a robust technique for breast cancer detection. The implications of our findings suggest a potential paradigm shift in early cancer detection, paving the way for enhanced patient care and management strategies.

## 5. Conclusions

In a promising pilot study involving 24 patients, our approach exhibited a remarkable 100% sensitivity and specificity rate in detecting all malignant breast tumors, without any false positive results observed in the contralateral breasts, irrespective of tumor size, location, and breast tissue density. Larger studies are in process to further illuminate the pathway for this new technology in breast cancer detection.

## Figures and Tables

**Figure 1 cancers-16-02264-f001:**
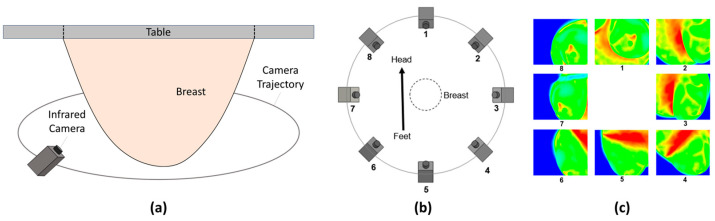
Illustration of infrared imaging procedure of (**a**) a breast under an imaging table in the prone position and (**b**) eight positions to extract multi-view infrared images. (**c**) Example of a patient’s infrared images obtained using infrared imaging in the prone position [52].

**Figure 2 cancers-16-02264-f002:**
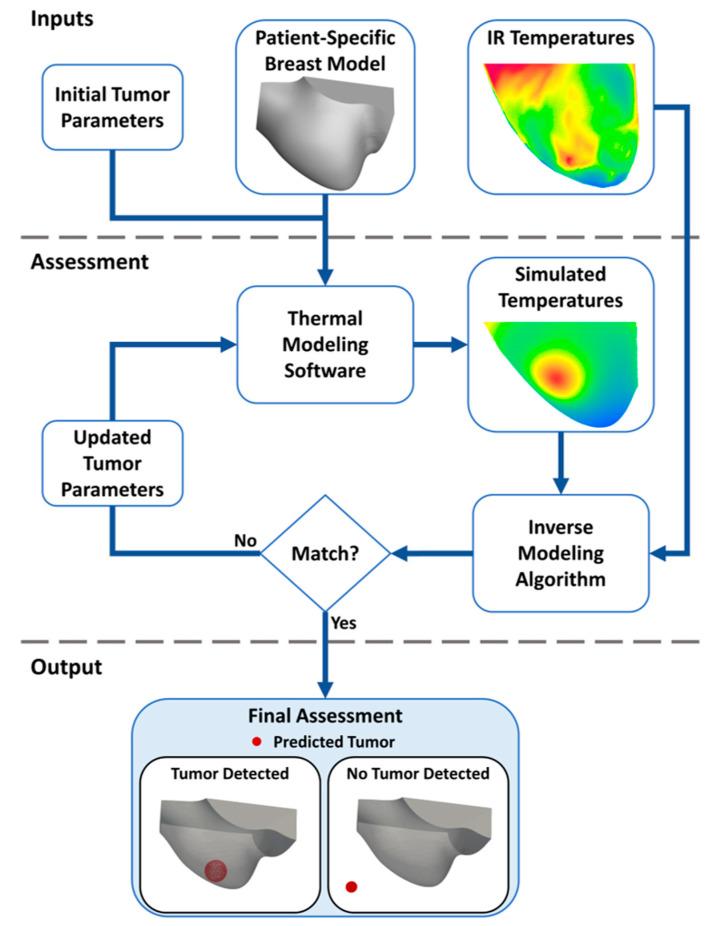
Flowchart of the developed breast cancer detection algorithm that utilizes thermal modeling and inverse modeling with patient-specific breast models and infrared surface temperatures. The inverse model provides a final assessment that indicates the presence or absence of a tumor heat source.

**Figure 3 cancers-16-02264-f003:**
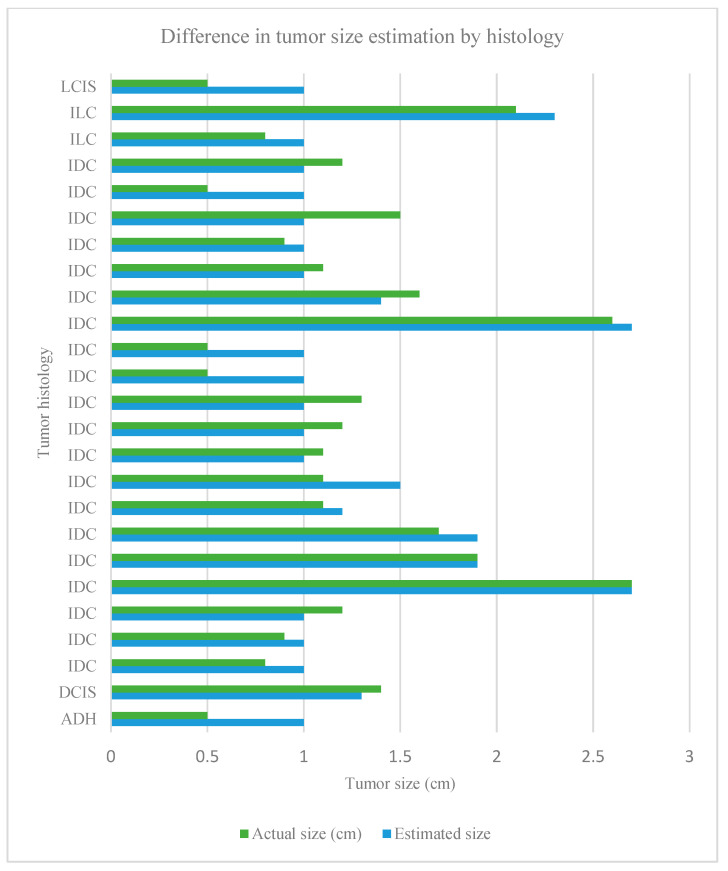
All 25 breast tumors with a comparison of the actual and estimated tumor size by histological subtype. Histology subtypes included invasive ductal carcinoma (IDC), invasive lobular carcinoma (ILC), ductal carcinoma in situ (DCIS), lobular carcinoma in situ (LCIS) and atypical ductal hyperplasia (ADH).

**Table 1 cancers-16-02264-t001:** Demographic and clinical data for the 25 malignant breast tumors: laterality designed as right (R) and left (L) location within the breast—upper outer quadrant (UOQ), upper inner quadrant (UIQ), lower outer quadrant (LOQ), and lower inner quadrant (LIQ); tumor grade indeterminate (X)—low (1), intermediate (2), and high-grade (3); presence of additional DCIS/ LCIS with invasive carcinoma; and TNM staging and the presence of prognostic markers including estrogen receptor (ER), progesterone receptor (PR), and HER2 receptor. Tumor depth was calculated using both the infrared images and MRI images.

Age	Laterality	Location	Breast Tissue Density	Histology	Grade	Actual Size (cm)	Estimated Size (cm)	DCIS/ LCIS	Staging	ER	PR	HER2	Depth of Tumor Based on MRI (cm)
68	L	UOQ	SF	ADH	X	0.5	1	None	—	—	—	—	2.91
60	R	UOQ	HD	DCIS	2	1.4	1.3	DCIS	T1c N0 Mx	+	+	—	1.18
71	R	UOQ	PF	IDC	1	0.9	1	None	T1c N0 Mx	+	-	1+	2.26
67	L	UIQ	ED	IDC	1	1.7	1.9	None	T1c N0 Mx	+	+	1+	2.4
67	L	UOQ	SF	IDC	1	1.1	1.2	None	T1c N0 Mx	+	+	2+	2.28
48	R	UOQ	SF	IDC	1	1.1	1	DCIS	T1c N0 Mx	+	+	1+	2.23
64	R	UOQ	SF	IDC	1	1.2	1	None	T1c N0 Mx	+	+	1+	4.95
65	L	UOQ	HD	IDC	1	1.6	1.4	None	T1c N0 Mx	+	+	1+	—
70	R	UOQ	SF	IDC	2	0.8	1	DCIS	T1c N0 Mx	+	+	1+	2.14
51	L	UOQ	SF	IDC	2	1.9	1.9	LCIS	T2 N1a M0	+	+	1+	0.95
46	R	LIQ	HD	IDC	2	1.1	1.5	DCIS	T1c N0 Mx	+	+	1+	2.96
72	R	UOQ	SF	IDC	2	1.1	1	DCIS	T1c N0 Mx	+	+	2+	2.74
64	L	UOQ	PF	IDC	2	1.5	1	DCIS	T1c N1a M0	+	+	2+	2.93
63	L	UIQ	SF	IDC	2	0.5	1	DCIS	T1b N0 Mx	+	+	0+	3.9
57	L	UIQ	SF	IDC	2	1.2	1	None	T1c N0 Mx	+	+	2+	2
52	R	UOQ	SF	IDC	3	1.2	1	DCIS	T1a N0 Mx	—	—	0+	5.45
68	R	UOQ	SF	IDC	3	2.7	2.7	DCIS	T2 N1a Mx	+	+	1+	2.72
68	L	UOQ	HD	IDC	3	1.3	1	None	T2 N0 Mx	—	—	0+	2.43
70	R	LIQ	SF	IDC	3	0.5	1	DCIS	T1b N0 Mx	—	—	0+	3.76
42	R	UOQ	HD	IDC	3	0.5	1	DCIS	T1b N0 Mx	—	—	0+	2.42
49	R	LOQ	SF	IDC	3	2.6	2.7	DCIS	T2 N0 Mx	—	—	0+	4.3
72	L	UIQ	SF	IDC	3	0.9	1	DCIS	T1b N0 Mx	—	—	0+	1.14
68	L	UOQ	SF	ILC	1	0.8	1	DCIS	T1c N1a M0	+	+	1+	2.93
70	L	UOQ	SF	ILC	2	2.1	2.3	LCIS	T2 N0 Mx	+	+	2+	2.27
67	L	UIQ	SF	LCIS	X	0.5	1	LCIS	—	—	—	—	1.29

## Data Availability

The original contributions presented in this study are included in the article; further inquiries can be directed to the corresponding author.

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
