# Peer review of "Breast Cancer Screening Using Inverse Modeling of Surface Temperatures and Steady-State Thermal Imaging"

_cancers, 2024, doi:10.3390/cancers16122264_

Round 1
Reviewer 1 Report
Comments and Suggestions for Authors
The paper is of interest and overall well-written. The authors are suggested to address few minor issues:
- The results could be enriched by adding TNM staging of the lesions to case description
- A sensitivity and specificity analysis should be carried out to improve study validity and assess the techniques's accuracy.
Author Response
Thank you for your valuable input and comments. The following are the responses to the reviewers’ comments and were edited in the manuscript.
Pg 5 table 1
TNM staging has been added to the table including all the demographic and clinical patient information.
Pg 7 Lines 241, 242
A sensitivity and specificity analysis were not included since this study was not conducted as a screening study involving the general population and all included patients had breast cancer, confirmed by biopsy. However, since all the tumors were accurately detected by the technique and no false positives noted in the contralateral normal breasts, a 100% sensitivity and 100% specificity has been included as a part of the conclusions

Reviewer 2 Report
Comments and Suggestions for Authors
The idea of non-ivasive method for analysis of breast tumor is of great potential for the practical application. Authors present the data on the experiments with group of patients demonstrating applicability of temperature mapping of breast and analyzing the results using modeling software. Despite the importance of the research the manuscript has not enough research data and its analysis to fit the research article requirements.
The details of the experiments (equipment, measurement parameters) are not fully described. No positive and negative control data is presented. No information about minimal size of the tumor (sensivity) that could be detected by the suggested method.
Authors describe group of women of age above 21 (line 8) but the minimal age of the patients in group is 42.
It is not clear how for the group of 24 women there were obtained 47 (25+22) results (see line 146)
Also, local change of the body tissues temperature could be result of different deseases. This factor should be discussed.
Author Response
The authors would like to thank the reviewer for their valuable feedback. We have revised the manuscript to include more details on the experiments, the positive and negative control data, and the minimal tumor sizing. The following was added to the manuscript:
Pg. 3 Lines 98-100
MRI images of bilateral breasts of the patients were captured in prone position with a GE 3T MRI scanner under an IRB approved protocol prior to biopsy [50]. Axial, coronal, and sagittal imaging were captured pre and post contrast. A figure (1) has also been included for visualization.
Pg. 5 Lines 161-162
In this work, the positive control was the breasts with verified malignancy, while the negative control was the contralateral healthy breasts.
Pg 6 Line 194
The sensitivity of tumor detection extended to tumors as small as 5 mm.
Pg. 7 Lines 220-223
Further studies on the minimal tumor size and other factors that affect the detectability of breast cancer using our algorithm are in development. Gutierrez and Kandlikar [54] have identified the detectability of the algorithm through numerical studies.
Authors describe group of women of age above 21 (line 8) but the minimal age of the patients in group is 42.
The inclusion criteria for the patients was age greater than age 21 years only. However, among the women enrolled in the study, the youngest patient was 42 years old.
It is not clear how for the group of 24 women there were obtained 47 (25+22) results (see line 146)
The number of contralateral breasts included were misnumbered and corrected as below.
Pg 5 Line 165
Final analysis included 24 patients diagnosed with breast tumors who underwent both MRI and infrared imaging, yielding a total of 25 breast tumors and 23 contralateral breasts devoid of tumors.
Pg 6 Line 191
Furthermore, none of the 23 contralateral breasts exhibited predicted tumors.
Also, local change of the body tissues temperature could be result of different deseases. This factor should be discussed
The authors would like to thank the reviewer for their valuable feedback. The algorithm models metabolically active and highly perfused malignant tumors using Pennes’ bioheat equation and Gautherie’s relation. These relations generate surface temperature that matches the temperatures captured in the IR images, which other diseases do not exemplify. For this reason, the algorithm is able ignore the local effects of caused by other factors such as thermal distribution caused by other diseases and even the local breast vascularity. This has been clarified in the manuscript and the following sentences have been added. This can be further supported by future studies that will aim to use this method for screening and help differentiate the heat signature from a tumor vs other etiologies.
Pg. 2 lines 56-59
Thermal imaging is a method that captures the infrared radiation emitted from the body captured by an infrared camera as surface temperature measurements [20]. In 1956, Lawson [21-24] showed that malignant tumors generated an increase in temperature measurements that can be captured through thermal imaging.
Pg. 2 lines 75-78
He developed a model that encapsulated the thermal effect from the metabolic activity as a heat source dependent on the tumor volume and doubling time [42-44]. This allowed researchers to model the tumor as a heat source dependent on the tumor size.
Pg. 7 lines 211-215
Our algorithm uses the concept that metabolically and highly perfused malignant tumors act as a heat source that affects the breast surface temperature. The algorithm consistently and effectively predicted breast cancer, irrespective of factors such as tumor location, breast tissue density, histological subtype, prognostic receptor status, or tumor depth using breast surface temperature measurements.

Reviewer 3 Report
Comments and Suggestions for Authors
This manuscript titled -Breast cancer screening using inverse modeling of surface temperatures and steady-state thermal imaging -is used a model which integrates a novel imaging technique and a physics-based model. The team wants to develop a new technique as a complementary tool to mammography for widespread screening. A total of 30 women were recruited for this study. Figure 2 in this paper is very effective and well prepared. The study design are interested and well performed. The algorithm demonstrated the potential for detection of all biopsy is making promises for a new diagnostics method. Quality of presentation can be improve by a one more figure.
The author should explain in the discussion section, why the number of references is small.
Thank you
Author Response
Thank you for your valuable input and comments. The references have been updated and an additional figure 1 has been added to the manuscript.

Round 2
Reviewer 2 Report
Comments and Suggestions for Authors
Authors have clarified the details of the investigation. So, the manuscript could be accepted for publication.